# Development of a Self-Harm Monitoring System for Victoria

**DOI:** 10.3390/ijerph17249385

**Published:** 2020-12-15

**Authors:** Jo Robinson, Katrina Witt, Michelle Lamblin, Matthew J. Spittal, Greg Carter, Karin Verspoor, Andrew Page, Gowri Rajaram, Vlada Rozova, Nicole T. M. Hill, Jane Pirkis, Caitlin Bleeker, Alex Pleban, Jonathan C. Knott

**Affiliations:** 1Orygen, Parkville, VIC 3052, Australia; katrina.witt@orygen.org.au (K.W.); michelle.lamblin@orygen.org.au (M.L.); gowri.rajaram@orygen.org.au (G.R.); Nicole.Hill@telethonkids.org.au (N.T.M.H.); caitlin.bleeker@orygen.org.au (C.B.); 2Centre for Youth Mental Health, The University of Melbourne, Parkville, VIC 3052, Australia; 3Centre for Mental Health, Melbourne School of Population and Global Health, The University of Melbourne, Parkville, VIC 3010 Australia; m.spittal@unimelb.edu.au (M.J.S.); j.pirkis@unimelb.edu.au (J.P.); 4Centre for Brain and Mental Health Research, Faculty of Health and Medicine, University of Newcastle, Callaghan, NSW 2308, Australia; Gregory.Carter@newcastle.edu.au; 5Calvary Mater Newcastle, Callaghan, NSW 2308, Australia; 6School of Computing and Information Systems, The University of Melbourne, Parkville, VIC 3052, Australia; karin.verspoor@unimelb.edu.au (K.V.); vlada.rozova@unimelb.edu.au (V.R.); 7Centre for Digital Transformation of Health, The University of Melbourne, Melbourne, VIC 3000, Australia; 8Translational Health Research Institute, Western Sydney University, Campbelltown, NSW 2560, Australia; A.Page@westernsydney.edu.au; 9Telethon Kids Institute, Nedlands, WA 6009, Australia; 10Mid-West Area Mental Health Service, Emergency Department, Sunshine Hospital, Sunshine, VIC 3021, Australia; alex.pleban@mh.org.au; 11Centre for Integrated Critical Care, Melbourne Medical School, The University of Melbourne, Parkville, VIC 3010, Australia; jonathan.knott@mh.org.au

**Keywords:** self-harm, suicide, monitoring, emergency department, artificial intelligence, natural language processing, Australia

## Abstract

The prevention of suicide and suicide-related behaviour are key policy priorities in Australia and internationally. The World Health Organization has recommended that member states develop self-harm surveillance systems as part of their suicide prevention efforts. This is also a priority under Australia’s Fifth National Mental Health and Suicide Prevention Plan. The aim of this paper is to describe the development of a state-based self-harm monitoring system in Victoria, Australia. In this system, data on all self-harm presentations are collected from eight hospital emergency departments in Victoria. A natural language processing classifier that uses machine learning to identify episodes of self-harm is currently being developed. This uses the free-text triage case notes, together with certain structured data fields, contained within the metadata of the incoming records. Post-processing is undertaken to identify primary mechanism of injury, substances consumed (including alcohol, illicit drugs and pharmaceutical preparations) and presence of psychiatric disorders. This system will ultimately leverage routinely collected data in combination with advanced artificial intelligence methods to support robust community-wide monitoring of self-harm. Once fully operational, this system will provide accurate and timely information on all presentations to participating emergency departments for self-harm, thereby providing a useful indicator for Australia’s suicide prevention efforts.

## 1. Introduction

In Australia, suicide accounted for 3046 deaths and was the leading cause of death among Australians aged 15–44 years in 2018 [1]. Self-harm (defined below) is more common [2]; for every death by suicide, it is estimated that there are over ten times the number of hospitalisations for self-harm [3,4]. As well as being problematic in its own right, self-harm is associated with a number of adverse outcomes, and is strongly associated with future suicide [2,5,6,7]. Monitoring instances of self-harm can serve as a useful indictor of the efficacy of suicide prevention efforts. For these reasons, the World Health Organization (WHO) has recommended that all member states develop a national or subnational self-harm surveillance system as part of their suicide prevention efforts [8].

The development of systems for the monitoring of non-fatal self-harm is a key priority under Australia’s Fifth National Mental Health and Suicide Prevention Plan [9,10] and funding has been provided by the Australian Government to develop a national suicide and self-harm monitoring system for Australia [11], currently being established by the Australian Institute of Health and Welfare. The aim of this system is to improve access to high-quality and timely data in order to inform policy initiatives and real-time (or close to real-time) responses and the intention is that data will feed into it from a range of sources, including emergency departments.

Emergency departments (EDs) are a common first point-of-contact for people who self-harm [12,13], and the period following discharge represents a time of elevated suicide risk. Approximately 15% of people who present to an ED with self-harm die by suicide within one year [14], although the risk of suicide can remain elevated for many years [15]. Therefore, EDs are not only a logical setting in which to locate a self-harm monitoring system, but also represent a critical opportunity for suicide intervention and prevention.

ED-based self-harm monitoring systems have been established around the world, including in the United States [16], Northern Ireland [17], the Republic of Ireland [18], France [19] and the United Kingdom [20,21,22]. Whilst they vary in scope, many have been used to inform the allocation of clinical resources, facilitate the identification of trends in the incidence of self-harm, and provide valuable epidemiological data for the evaluation of public health interventions aimed at reducing self-harm and suicide at a population level [23]. These existing systems provide important information that can be used in the establishment of a system for Australia, in particular with regard to accuracy of case identification and timeliness.

For example, one key challenge for many existing systems is the reliance on ICD-10 and/or diagnosis codes to identify self-harm cases [24,25]. These codes are often used to identify health outcomes from clinical records like liver disease, diabetes and certain cancers, with a moderate-to-high level of sensitivity [26]. However, in the absence of additional data these codes lack the sensitivity required to accurately classify self-harm presentations [27]; for example, they are unable to distinguish between an accidental overdose and one attributable to self-harm [23]. As such, systems based on ICD-10 and/or diagnosis codes alone are likely to under-enumerate cases of self-harm [16,27,28,29].

A further consideration relates to timeliness. Most of the international systems mentioned above currently rely on manual data extraction and coding processes, which adversely affects their timeliness and renders them inadequate for real-time (or even close to real-time) monitoring [23]. From a public health perspective, delays in monitoring can lead to missed opportunities for urgent responses; for example, to suspected clusters of self-harm or suicide as they emerge.

Whilst EDs Australia-wide collect information on injury intent, including by self-harm, they are reliant on the completion of additional data capture forms by busy triage staff. This places considerable time burdens on staff and as a result such information is often incomplete or inaccurate. To date only one comprehensive ED-based self-harm monitoring system currently exists in Australia. The Hunter Area Toxicology Service monitors drug overdoses and self-poisoning for one health catchment area in Newcastle, Australia [30]. It serves a population of over 410,000 (Newcastle, Lake Macquarie and Port Stephens) and is also a tertiary referral centre for a further rural population of over 243,000 (Lower and Upper Hunter Valley) [31]. However, because it only captures episodes of intentional drug overdose and/or self-poisoning and excludes other methods of self-harm, this system is limited in its capacity to provide comprehensive self-harm monitoring. Two other systems are also in development, one in Sydney, New South Wales (capitalising on an existing study of individuals presenting to ED following self-harm [32]), and one on the Gold Coast in Queensland; however, to the best of our knowledge, neither were fully operational at the time of writing.

Thus, there is the need for a robust, regionally-based self-harm monitoring system in Victoria in order to (1) estimate the prevalence of self-harm presentations to EDs across the state of Victoria (including an examination of trends over time); (2) describe the demographic, clinical and treatment characteristics of people who present; (3) inform real-time (or close to real-time) responses; and (4) inform the development, and evaluation, of clinical and policy initiatives.

## 2. Objective

The objectives of this paper are to describe the development of the Self-harm Monitoring System for Victoria, Australia, including the geographical sites included, the data collected, the natural language processing classifier to be applied, and how the system will be used to improve data collection and clinical practice.

## 3. Materials and Methods

### 3.1. Setting

The development of this system is being led by researchers at Orygen and the Centre for Youth Mental Health, The University of Melbourne, in partnership with our ED site colleagues. It is being developed in eight (of 38) public hospitals with 24-h EDs across the state of Victoria, six of which are located in metropolitan Melbourne and two in regional Victoria (see Figure 1). Whilst these sites are broadly representative of the Victorian population, and will capture almost three quarters of self-harm presentations to EDs [33], there may be an over-representation of persons from culturally and linguistic diverse (CALD) communities and an under-representation of people who identify as Aboriginal or Torres Strait Islander [34]—see Figure 2 and Figure 3.

Ethical approval has been granted by the Melbourne Health Human Research Ethics Committee (HREC; 2017.342).

### 3.2. Data Sources

De-identified data on all presentations since 1 January 2012 across the eight participating sites are currently provided to the research team as quarterly exports via secure exchange server. These are automatically mapped and loaded into a standardised study database where a series of pre-acceptance validation checks are undertaken. These checks ensure that any sensitive data fields (e.g., names and addresses) are automatically removed. As each hospital site uses different patient management systems (which can differ over time, for example if a site transitions to a new patient data management system), these checks also ensure that the format of the dataset is harmonised between sites.

Data collected include information on patient characteristics (e.g., unit record (UR) number, sex/gender, age, postcode), circumstances of the presentation (e.g., date and time, presentation complaint, free-text triage comments, ICD-10 code, diagnostic codes, circumstances of the injury if relevant), information on any management received in the ED (e.g., time seen by a doctor and/or emergency mental health clinicians as necessary) and disposition (e.g., discharge date and time, discharge destination)—see Figure 4. Further post-processing is then undertaken to ensure unique patient identifiers are replaced with a mapped study-specific identifier.

### 3.3. Inclusion and Exclusion Criteria

Consistent with WHO and other international definitions [21,22,35,36,37], self-harm is defined as “an act with a non-fatal outcome in which an individual deliberately initiates a non-habitual behaviour that, without intervention from others, will cause harm to the self, or [when an individual] deliberately ingests a substance in excess of the prescribed or generally recognised therapeutic dosage” [35]. The term ”deliberately” distinguishes self-harm from non-intentional injury and poisoning. Due to high rates of co-occurrence and difficulties establishing intent, there is no distinction between attempted suicide (i.e., self-harm where there is evidence of suicidal intent) and non-suicidal self-injury (i.e., where there is no evidence of suicidal intent) [38,39]. All presentations for self-harm involving the use of any method are included. Cases where the ED presentation did not represent the first help-seeking occasion for that episode (i.e., where the patient was seeking help for the delayed consequences of a previous self-harm episode) are excluded, as are cases where the patient died prior to arrival (i.e., suicide cases).

### 3.4. Natural Language Processing for Case Ascertainment

In order to overcome the challenges inherent in identifying self-harm cases from structured data fields alone [40,41], a natural language processing (NLP) classifier is being developed to identify cases of self-harm from free-text triage case notes. To build this classifier, we firstly manually coded all presentations from Royal Melbourne Hospital (N episodes = 497,480) over a five-year period (2012–2017).

Next, the data were split into a training/development subset and a held-out (test) subset in an 80/20 proportion. The development set was used to experiment with different machine learning approaches and to optimise hyperparameters, using a five-fold cross-validation training/evaluation paradigm. The final model was tested on the held-out (test) data to assess the performance on unseen data. ED notes are rapidly written short texts, with heavy use of clinical concepts and abbreviations and frequent grammatical and spelling mistakes, thereby posing a challenge to traditional NLP methods [42]. Therefore, pre-processing is often required for data normalisation. To correct spelling, a custom dictionary including general English, medical terms, drug names and domain-specific abbreviations is being developed. Another common issue when working with textual data is the extreme sparsity of the vocabulary feature space (i.e., a particular term might occur in the whole corpus only a handful of times). Most machine learning models are not very effective at handling highly sparse matrices; thus, feature selection techniques should be applied to reduce the dimensionality of the problem.

An initial model based on the support vector machine (SVM) learning algorithm was associated with high levels of precision, or positive predictive value (>0.70), and acceptable levels of recall, or sensitivity (>0.55), when based on free-text triage case note information alone. We continue to experiment with strategies to improve this performance by refining the pre-processing steps and testing other modelling approaches. For example, word and document embedding techniques allow capturing the semantic and syntactic context of a word to improve the predictive power of the model [43]. Alternatively, topic modelling, which estimates the set of topics represented in each document, has been shown to improve the sensitivity of a triage note classification system [44]. Furthermore, recurrent neural network models, in particular long short-term memory architectures, are capable of learning from sequential data and are becoming increasingly popular in NLP tasks. In the context of ED notes, several recent studies experimenting with these models have shown promising results in classifying unstructured free-text data compared to traditional machine learning [45,46]. We will describe the full technical details of our experimentation in a future publication focused on the NLP methods.

### 3.5. Post-Processing

A combination of data fields, including, but not limited to, ICD-10 codes, diagnosis codes and free text triage notes, are subject to post-processing to identify primary mechanism of injury and specific medications consumed. Primary mechanism of injury is coded according to the following ICD-10 Australian Modification (ICD-10-AM) [47] intentional injury codes: overdoses of drugs and medicaments (X60–64); self-poisoning by alcohol (X65); poisonings involving the ingestion of chemicals, noxious substances, gases or vapours (X66–69); hanging, strangulation and suffocation (X70); drowning and submersion (X71); use of firearms or explosives (X72–74); sharp objects (X78); falls from a height (X80); and ”other” methods (X75–77, X79, X81–84). Where multiple methods are used, we code for the method associated with the greatest potential lethality as reflected in published case fatality ratio data from Australia [48].

As above, information on the specific medication preparation(s) consumed is coded according to the WHO’s Anatomical Therapeutic Chemical (ATC) classification system [49] and illicitly-sourced drugs are defined as those listed in Schedule 9 or Schedule 8 of the Poisons Standard (2020) [50]. The number of tablets consumed is also recorded where reported.

Finally, the presence of psychiatric disorders is captured from free text fields (where available).

### 3.6. Quality

To ensure both the accuracy and consistency of the case ascertainment processes, a number of quality assurance activities are conducted. Firstly, at each site periodic manual file audits of a selection of cases are conducted to ensure the data received from each ED are sufficient to make the determination of self-harm accurately. Results from these analyses are used to determine whether additional data fields should feed into the NLP classifier in order to refine the process. Second, as many machine learning-based classifiers assign a confidence score for each decision (noting that optimising the confidence threshold is part of the model tuning process), a cross-checking process is undertaken for all predictions associated with low confidence, hence flagged as unclear. A pair of research assistants (GR and CB) independently review these cases and consensus discussions with the research fellow (KW) are conducted to resolve cases requiring a decision on whether the case should be classified as self-harm.

### 3.7. Data Analysis

Data will be analysed in order to address the broad aims/research questions outlined above, namely to examine the prevalence of presentations to ED for self-harm (including examining trends over time and in relation to specific events), examine the demographic, clinical and treatment characteristics of those who present, inform real-time (or close to real-time) responses to self-harm and inform the development, and evaluation, of clinical and policy initiatives.

### 3.8. Prevalence and Characteristics of Self-Harm Presentations

To determine the prevalence of presentations for self-harm to each of our eight ED sites, age-standardised rates will be calculated. Age-standardised rates per males, females and intersex patients will also be calculated to enable examination of any changes in presentation rates by sex/gender over time.

### 3.9. Characterising the Demographic, Clinical and Treatment Characteristics of Those Who Present to EDs Following Self-Harm

To characterise the demographic, clinical and treatment characteristics of those who present to EDs following an episode of self-harm, a series of regression models will be implemented. These models will enable us to examine whether certain patient (e.g., patient demographics, such as age or sex/gender), clinical (e.g., psychiatric diagnoses) and/or presentation (e.g., self-harm method used, presence/absence of alcohol intoxication) factors affect the treatment patients receive in the ED. Data linkage will also be undertaken to explore and capture diversity in the treatment patients presenting to these EDs receive in the post-discharge period and, additionally, whether any of these factors (i.e., patient, clinical and/or presentation) influences these treatment pathways.

### 3.10. Informing Real-Time (or Close to Real-Time) Responses to Self-Harm

By providing rich data on the prevalence of self-harm and the demographic, clinical and treatment characteristics of those who present to EDs following an episode of self-harm, outputs from our system will also be able to inform real-time (or close to real-time) responses to self-harm. For example, as our system captures information on postcode, data from our system could be used to geographically map presentations to aide in the identification of emergent spatial and/or temporal self-harm clusters. Working together with community organisations, we could then partner to develop a community response to these clusters, the effectiveness of which we could then monitor using our system. Given that we also capture information on the self-harm method used at each presentation, our system could also be used to identify emerging new self-harm methods in a timely way.

### 3.11. Inform the Development, and Evaluation, of Clinical and Policy Initiatives

Finally, data from our system can be leveraged by clinical practitioners and policymakers to develop, monitor, refine and evaluate initiatives. For example, throughout Victoria, the Hospital Outreach Post-suicidal Engagement (HOPE) initiative is currently being rolled out for adults (aged 18 years or older) presenting to EDs following an episode of self-harm and/or acute suicidal ideation who do not meet clinical thresholds for in- and/or out-patient psychiatric treatment. A further initiative is also currently under development for children and adolescents up to 18 years of age. Data from the Victorian Self-Harm Monitoring System can be used to monitor the ongoing effectiveness of these initiatives at the population level, whilst providing valuable information as to which component(s) of these complex multicomponent initiatives may be most effective.

## 4. Discussion

The Self-Harm Monitoring System for Victoria will provide up-to-date and accurate information on all presentations to participating EDs for self-harm. Despite improved monitoring of self-harm being a national policy priority [10], to date no robust state-based system for monitoring self-harm presentations exists across Australia. This system is being developed in line with international best practice [35] and captures a standardised set of variables, which will ultimately allow for harmonisation with systems being developed in other jurisdictions as they come into full operation.

The system will make it possible to answer a range of emerging research questions relating to the epidemiology of self-harm presentations to EDs and including changes in rates of presentation over time and following significant events. For example, it will be possible to shed light on whether or not rates of presentation changed during, and immediately after, the COVID-19 pandemic, or whether or not rates of presentation increase following significant media depictions of self-harm or suicide, such as in the case of the Netflix series *13 Reasons Why*, as reported internationally [51,52]. Additionally, the system will be used to forecast trends in self-harm to inform resource planning and service capacity needs in the short and medium term. It will also be able to provide information on the demographic and clinical characteristics of people who present, on the treatment people receive and on rates of re-presentation, which taken together have the potential to help inform clinical practice and policy development going forward.

Similarly, the system will also have the capacity to evaluate changes in clinical practice and the impact of key policy initiatives over time. For example, a similar system in the UK has been used to monitor the emergence of novel methods of self-harm [53] and to evaluate the impact of policy initiatives such as restricting the availability of paracetamol and co-proxamol on presentations for overdose by that method [54,55].

In Victoria, a cornerstone of the current suicide prevention efforts has been the development of a number of “aftercare services” designed to provide support to people for up to three months following hospital presentation for suicide risk [56]. The Self-Harm Monitoring System for Victoria will be able to examine rates of self-harm presentations before and after the implementation of these services, thereby providing valuable data to both individual services and the state government on the likely efficacy of these initiatives

### Strengths and Limitations

A key strength of the Self-Harm Monitoring System for Victoria is that, unlike existing systems in Australia, information on all self-harm presentations will be captured in detail. The system also has excellent geographic coverage, networking eight hospital catchment regions and representing 3,342,315 people, or 51% of Victoria’s population as of 1 July 2019.

Additionally, the system does not impose any additional coding or clerical burdens on clinical staff. This is important as greater data collection burden has been negatively associated with timeliness for some similar systems internationally [23]. Instead, the Self-Harm Monitoring System for Victoria leverages data that are collected for routine clinical, rather than research, purposes. This may negatively affect the accuracy and comprehensiveness of variables that do not directly affect clinical decision-making or management; for example, self-reported lifetime histories of self-harm or psychiatric diagnoses [57]. Anecdotal evidence suggests that implementation of the system itself has already led to improved coding practices within some participating EDs and we will provide participating sites with ongoing training and professional development through their engagement with the system.

A further strength is the use of specifically developed machine learning algorithms which will expedite the case identification process, hence enabling us to process data far more quickly than some older systems [19]. These algorithms are still under development but show significant promise for improving robust identification of relevant cases of self-harm. Thus, as the system in its current form only collects data quarterly (making it comparable to international systems [18,22]), this renders it inadequate for real-time monitoring in its current form. However, the application of the machine learning algorithms, together with plans to increase the frequency of data collection once the system is fully operational, mean that the system will not just be used for research and evaluation purposes but also to inform real-time responses.

A final limitation is that at present the system does not allow for data linkage to other administrative databases, such as mortality data. Australia, unlike some jurisdictions internationally, does not have a personal identity number system for the identification of individuals and, until recently, no purpose-built national data linkage infrastructure [58]. As a consequence, research involving linked data in Australia has typically been complex, time-consuming and costly to undertake [59]. However, this will be addressed in future iterations of the system.

## 5. Conclusions

Notwithstanding the limitations, the Self-Harm Monitoring System for Victoria will provide a critical piece of infrastructure that will deliver robust, ongoing and up-to-date information on all ED presentations for self-harm. It will have the capacity to both inform and evaluate future clinical practice and policy development in Victoria. The system has the capacity to be further expanded across Australia, which would not only align with Australia’s current suicide prevention policy [10] but would also bring Australia in line with international best practice recommendations.

## Figures and Tables

**Figure 1 ijerph-17-09385-f001:**
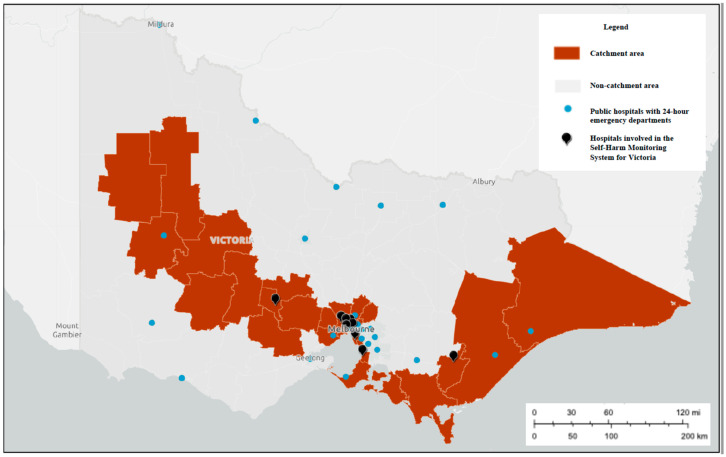
Geographical location of eight emergency departments (EDs) currently participating in the Self-Harm Monitoring System for Victoria.

**Figure 2 ijerph-17-09385-f002:**
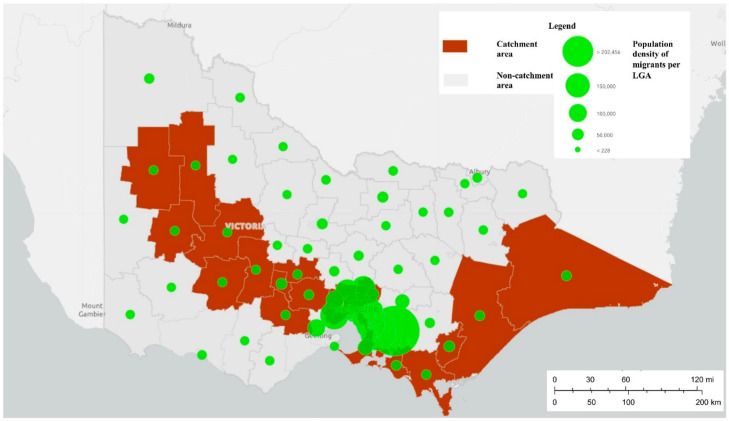
Population density per local government area (LGA) of migrants from culturally and linguistically diverse (CALD) communities in the catchment areas of the Self-Harm Monitoring System for Victoria.

**Figure 3 ijerph-17-09385-f003:**
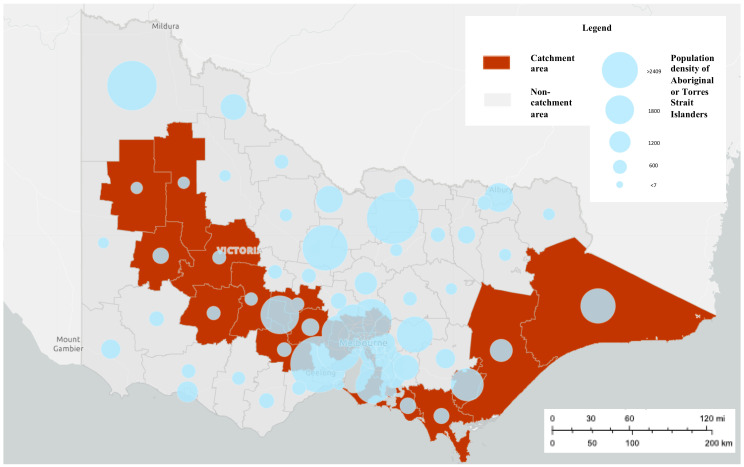
Population density of persons from Aboriginal or Torres Strait Islander backgrounds in the catchment areas of the Self-Harm Monitoring System for Victoria.

**Figure 4 ijerph-17-09385-f004:**
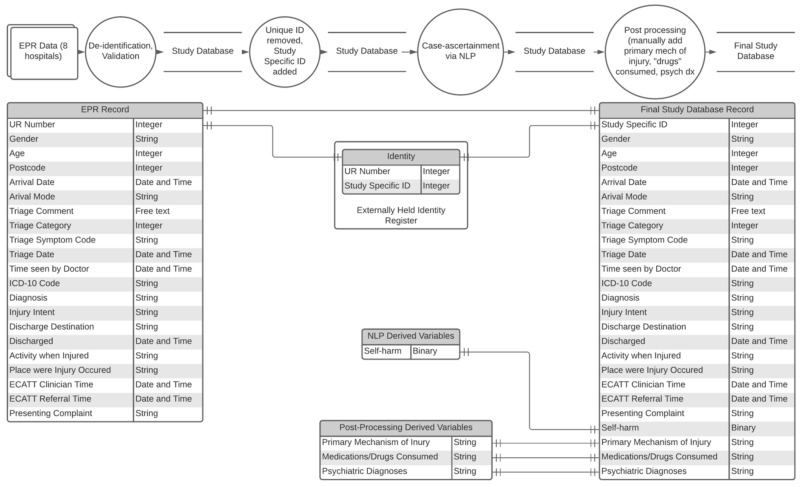
Data flow diagram of the Self-Harm Monitoring System for Victoria. Notes: dx: diagnosis; ECATT: Enhanced Crisis Assessment Treatment Team; EPR: electronic patient record; ICD-10: International Classification of Disease, version 10; ID: identity; NLP: natural language processing; UR: unit record.

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
