# Peer review of "Development of a Self-Harm Monitoring System for Victoria"

_ijerph, 2020, doi:10.3390/ijerph17249385_

Round 1

Reviewer 1 Report

This study sought to develop a regional/national self-harm monitoring system to report cases and eventually prevent self-harm and suicide. The presentation is sound and detailed. The manuscript is well-written, and methods are clear and appropriate. The only recommendation is to consider revising the title. It would be helpful to clarify that the Victoria from "Victorian" self-harm monitoring system is referring to a geographical location- Victoria, Australia. 

Author Response

Many thanks for this suggestion. We have now amended the title to read: Development of a Self-harm Monitoring System for Victoria, Australia.

We have also referred to the system as "The Self-Harm Monitoring System for Victoria" throughout the manuscript. we hope this is now clearer.

Reviewer 2 Report

This article is not properly a scientific article but rather a summary of a project proposal.

In particular, the work done to date consisted of creating a process for obtaining normalized data from various sources and choosing a machine learning classifier (based on SVM). Results are not presented as they have not yet been obtained.

I consider the project is very interesting and that when data on the results be obtained, a very interesting paper will be published but that moment has not yet come.

As far as the classifier is concerned, much more information needs to be provided:

* Information on the size and composition of the training and test datasets.

* Information on the configuration of each of the models (approaches) that have been tested

* Results of each of the models tested and reasons to choose SVM as classification model.

It would also be interesting to do an analysis of the most relevant characteristics for  the chosen model, for example using some measure such as "information gain" (most of the automatic classification toolkits provide some mechanism to obtain the information gain associated to each feature).

In lines 165-174 it is indicated that the dataset was divided into a part of training and another of test (it is called “development” but it is for testing, actually). It is also indicated the use of cross-validation, without indicating exactly what for, we assume that it will be for parameter tuning, which is what it is usually used for.

When the dataset is large enough, the usual practice is to split it into three sections:

* one for training (training set)

* one for parameter setting (development set)

* one for test (testing set)

Certainly, if it is not possible to get a development set, it can be substituted by 5-fold or 10-fold cross-validation on the training set

Other remarks:

Figure captions are missing.

Author Response

We thank the reviewer for this comment. However, it was our deliberate intention to present a descriptive paper that detailed the development of the system and how we intend for it to be used. As the reviewer will be aware there is an increasing move towards Open Science and the publication of study protocols across academia, and including in suicide prevention (Pirkis, 2020), hence our rationale for publishing the manuscript in its current form.

We also note that when we were developing this system it would have been extremely helpful to have access to the protocols of similar systems that exist around the world; as it was we had to rely on broad guidance and personal connections to ensure that our system was aligned with international best practice and compatible with similar systems elsewhere. We very much hope that this descriptive, protocol-style paper will be of use to others who may wish to develop similar systems in the future.

We have however tried to address some of the concerns expressed by providing some more information, as detailed below. We have also amended the Aims to make our intentions clearer.

  1. As far as the classifier is concerned, much more information needs to be provided:
  • Information on the size and composition of the training and test datasets.
  • Information on the configuration of each of the models (approaches) that have been tested
  • Results of each of the models tested and reasons to choose SVM as classification model.

Thank you for your comments.

The size of the data set is over 400k episodes as indicated in the section describing the NLP (p4-5). We have clarified the size of the data splits in our revision; 80% for training and tuning, and 20% as a held-out test set. We report results on an initial system based on SVM that we have developed in order to assess the promise of the approach. Further experimentation of the models is ongoing, as we state (e.g., Abstract). A separate paper focusing on this NLP experimentation is in development; however, it is out of scope for this paper, given the primary aim of presenting a description of the development of the overall monitoring system, rather than a focus purely on the NLP component.  

  1. It would also be interesting to do an analysis of the most relevant characteristics for the chosen model, for example using some measure such as "information gain" (most of the automatic classification toolkits provide some mechanism to obtain the information gain associated to each feature)

Indeed, this is part of our ongoing analysis and we look forward to presenting it in a more detailed study (as noted above). Please note that given the high dimensionality of the feature space for this data precludes individual feature-level information gain analysis for all features; we are more interested in understanding the impact of higher-level pre-processing, representational and modelling choices. We have added further discussion of these to the manuscript.

  1. In lines 165-174 it is indicated that the dataset was divided into a part of training and another of test (it is called “development” but it is for testing, actually). It is also indicated the use of cross-validation, without indicating exactly what for, we assume that it will be for parameter tuning, which is what it is usually used for. When the dataset is large enough, the usual practice is to split it into three sections:
    • one for training (training set)
    • one for parameter setting (development set)
    • one for test (testing set)

Certainly, if it is not possible to get a development set, it can be substituted by 5-fold or 10-fold cross-validation on the training set.

As stated in the NLP section (p4-5), we split the data set into development and test subsets. That paragraph states that the development set is used for both training and tuning under a 5-fold cross-validation paradigm. We prefer that to a fixed training/dev (tuning)/test split as it allows us to assess the robustness of the model across different data samples; in this way we perform the training/tuning cycle multiple times utilising the development set rather than just once. The test set is then maintained exclusively to evaluate performance of the tuned model on previously unseen data.

We have clarified this description in the manuscript to avoid confusion as follows:

“Next, the data were split into a training/development subset and a held-out (test) subset in an 80/20 proportion. The development set was used toexperiment with different machine learning approaches and to optimise hyperparameters, usinga five-fold cross-validation training/evaluationparadigm. The final model is tested on theheld-out (test)data to assess the performance on unseen data.” (p4-5).

  1. Figure captions are missing.

Thank you; this has been fixed.

Reviewer 3 Report

This paper reports how the authors built the Victorian Self-harm Monitoring System. Overall, the paper is well written. However, I fail to see its research value. What is the researching finding? Unlike references [50-52], the current paper only collected data but did not analyze the data to discover any phenomenon. So, the contribution of this paper should be the framework of the system, I guess. However, I don’t see any novelty of the entire system framework. Although the authors did use machine learning and NLP methods to classify the data, this does not seem to be the core of the paper because the adopted methods were not new and were not described in details in the paper.

Other comments:

Add numbering and caption for each figure and each table. The legend at the top right corner of Fig. 1 seems truncated.

Where is Table 1?

Author Response

We thank the reviewer for this comment. However, as noted above, it was our deliberate intention to present a descriptive paper that detailed the development of the system and how we intend for it to be used. As the reviewer will be aware there is an increasing move towards Open Science and the publication of study protocols across academia, including in suicide prevention (Pirkis, 2020), hence our rationale for publishing the manuscript in its current form.

We also note that when we were developing this system it would have been extremely helpful to have access to the protocols of similar systems that exist around the world; as it was we had to rely on broad guidance and personal connections to ensure that our system was aligned with international best practice and compatible with similar systems elsewhere. We very much hope that this descriptive, protocol-style paper will be of use to others who may wish to develop similar systems in the future.

We have amended the Aims to make our intentions clearer. However, we have also provided additional information about the NLP methods which we hope will address some of the concerns expressed. Please see pages 10-11.

  1. Add numbering and caption for each figure and each table. The legend at the top right corner of Fig. 1 seems truncated.

Thank you; this has been fixed.

  1. Where is Table 1?

Apologies, this is an oversight from a previous version of the manuscript. The information that was contained in Table 1 was incorporated into Figure 4. We have now deleted reference to this table in the manuscript.

Reviewer 4 Report

This paper describes the development of a state-based self-harm monitoring system. This system will leverage routinely collected data in combination with advanced artificial intelligence methods to support robust community-wide monitoring of self-harm.

The aim of this system is to improve access to high quality and timely data in order to inform policy initiatives and real-time (or close-to real time) responses and the intention is that data will feed into it from a range of sources, including emergency departments.

The work would be enriched with a more detailed description of artificial intelligence methods, machine learning, mapped study and series of regression models that will be implementedIn order to characterise the demographic, clinical, and treatment characterise of those who present to EDs following an episode of self-harm.  

Author Response

Thank you for your comment. We have added further discussion of the challenges of NLP over the ED texts to the manuscript, and outlined the approaches we are currently exploring to improve the modelling, as follows:

“ED notes are rapidly written short texts, with heavy use of clinical concepts and abbreviations, and frequent grammatic and spelling mistakes, thereby posing a challenge to traditional NLP methods{Kocbek, 2014 #55}. Therefore, pre-processing is often required for data normalisation. To correct spelling, a custom dictionary including general English, medical terms, drug names, and domain-specific abbreviations is being developed. Another common issue when working with textual data is the extreme sparsity of the vocabulary feature space (i.e., a particular term might occur in the whole corpus only a handful of times). Most machine learning models are not very effective at handling highly sparce matrices, thus, feature selection techniques should be applied to reduce the dimensionality of the problem.

An initial model based on the Support Vector Machine (SVM) learning algorithm was associated with high levels of precision, or positive predictive value (>0.70) and acceptable levels of recall, or sensitivity (>0.55) when based on free-text triage case note information alone. We continue to experiment with strategies to improve this performance by refining the pre-processing steps and testing other modelling approaches. For example, word and document embedding techniques allow capturing the semantic and syntactic context of a word to improve the predictive power of the model {Sterling, 2019 #56}. Alternatively, topic modelling, which estimates the set of topics represented in each document, has been shown to improve the sensitivity of a triage note classification system {Horng, 2017 #57}. Furthermore, recurrent neural network models, in particular long short-term memory architectures, are capable of learning from sequential data and are becoming increasingly popular in NLP tasks. In the context of ED notes, several recent studies experimenting with these models have shown promising results in classifying unstructured free-text data compared to traditional machine learning {Gilgorijevic, 2018 #58}{Vu, 2019 #59}. We will describe the full technical details of our experimentation in a future publication focused on the NLP methods.”(pp. 4-5).

Round 2

Reviewer 2 Report

The authors have corrected some of the deficiencies indicated in the review of the first version. However, I consider that the main objection of the first version remains: This article is not properly a scientific article but rather a summary of a project proposal (i.e., a description of future work).

In their response, authors said that “We very much hope that this descriptive, protocol-style paper will be of use to others who may wish to develop similar systems in the future”. However, we do not know if the protocol described in this article is useful until we get results from the system to be implemented. I hope that the system will be useful and will obtain good results. At that time  will be the moment to publish an article describing the protocols used in the design of the system, the results obtained and the problems encountered.

Reviewer 3 Report

The authors have added new content to clarify the objective of their research. The description of the experiment method is also improved in this revision. Although the technical contribution is weak, this research provides an interesting case study. I thus recommend the acceptance of the manuscript for publication.